

# A reference cytochrome c oxidase subunit I database curated for hierarchical classification of arthropod metabarcoding data

Rodney T. Richardson[1], Johan Bengtsson-Palme[2,3], Mary M. Gardiner[1] and Reed M. Johnson[4]

[1] Department of Entomology, Ohio State University, Columbus, OH, United States of America
[2] Department of Infectious Diseases, Institute of Biomedicine, The Sahlgrenska Academy, University of Gothenburg, Gothenburg, Sweden
[3] Center for Antibiotic Resistance Research (CARe), University of Gothenburg, Gothenburg, Sweden
[4] Department of Entomology, Ohio State University, Wooster, OH, United States of America

Corresponding author
Rodney T. Richardson,
richardson.827@osu.edu

## ABSTRACT

Metabarcoding is a popular application which warrants continued methods optimization. To maximize barcoding inferences, hierarchy-based sequence classification methods are increasingly common. We present methods for the construction and curation of a database designed for hierarchical classification of a 157 bp barcoding region of the arthropod cytochrome c oxidase subunit I (COI) locus. We produced a comprehensive arthropod COI amplicon dataset including annotated arthropod COI sequences and COI sequences extracted from arthropod whole mitochondrion genomes, the latter of which provided the only source of representation for Zoraptera, Callipodida and Holothyrida. The database contains extracted sequences of the target amplicon from all major arthropod clades, including all insect orders, all arthropod classes and Onychophora, Tardigrada and Mollusca outgroups. During curation, we extracted the COI region of interest from approximately 81 percent of the input sequences, corresponding to 73 percent of the genus-level diversity found in the input data. Further, our analysis revealed a high degree of sequence redundancy within the NCBI nucleotide database, with a mean of approximately 11 sequence entries per species in the input data. The curated, low-redundancy database is included in the Metaxa2 sequence classification software (http://microbiology.se/software/metaxa2/). Using this database with the Metaxa2 classifier, we performed a cross-validation analysis to characterize the relationship between the Metaxa2 reliability score, an estimate of classification confidence, and classification error probability. We used this analysis to select a reliability score threshold which minimized error. We then estimated classification sensitivity, false discovery rate and overclassification, the propensity to classify sequences from taxa not represented in the reference database. Our work will help researchers design and evaluate classification databases and conduct metabarcoding on arthropods and alternate taxa.

## INTRODUCTION

With the increasing availability of high-throughput DNA sequencing, scientists with a wide diversity of backgrounds and interests are increasingly utilizing this technology to achieve a variety of goals. One growing area of interest involves the use of metabarcoding, or amplicon sequencing, for biomonitoring, biodiversity assessment and community composition inference (*Yu et al., 2012*; *Guardiola et al., 2015*; *Richardson et al., 2015*). Using universal primers designed to amplify conserved genomic regions across a broad diversity of taxonomic groups of interest, researchers are afforded the opportunity to survey biological communities at previously unprecedented scales. While such advancements hold great promise for improving our knowledge of the biological world, they also represent new challenges to the scientific community.

Given that bioinformatic methods for taxonomic inference of metabarcoding sequence data are relatively new, the development, validation and refinement of appropriate analytical methods is ongoing. Relatively few studies have characterized the strengths and weaknesses of different bioinformatic sequence classification protocols (*Porter & Golding, 2012*; *Bengtsson-Palme et al., 2015*; *Peabody et al., 2015*; *Somervuo et al., 2016*; *Richardson, Bengtsson-Palme & Johnson, 2017*). Further, researchers continue to utilize a diversity of methods to draw taxonomic inferences from amplicon sequence data. Relative to alignment-based nearest-neighbor and lowest common ancestor-type classification approaches, methods involving hierarchical classification of DNA sequences are popular as they are often designed to estimate the probabilistic confidence of taxonomic inferences at each taxonomic rank. However, studies explicitly examining the accuracy of classification confidence estimates are rare (*Somervuo et al., 2016*).

When performing hierarchical classification, the construction, curation and uniform taxonomic annotation of the reference sequence database is an important methodological consideration. Database quality can affect classification performance in numerous ways. For example, artifacts within the taxonomic identifiers of a reference database can represent artificial diversity and the inclusion of sequence data adjacent to the exact barcoding locus of interest likely display sequence composition that is unrepresentative of the barcoding locus. Lastly, sequence redundancy within reference databases increases computational resource use and is particularly problematic for classification software programs that classify sequences based on a set number of top alignments. In general, such database artifacts have the potential to bias model selection and confidence estimation both with k-mer style classifiers such as UTAX, SINTAX and the RDP Naïve Bayesian Classifier (*Wang et al., 2007*; *Edgar, 2015*; *Edgar, 2016*) and alignment-based classification approaches such as Metaxa2 and Megan (*Huson et al., 2011*; *Bengtsson-Palme et al., 2015*). Thus, it is important to identify and manage reference sequence database artifacts during curation for optimal downstream classification performance.

The use of molecular barcoding and metabarcoding in arthropod community assessment and gut content analysis has gained popularity in recent years (*Corse et al., 2010*; *Yu et al., 2012*; *Mollot et al., 2014*; *Elbrecht & Leese, 2017*). However, as with other non-microbial taxonomic groups of interest, few researchers have developed hierarchical DNA sequence

classification techniques for arthropods (*Porter et al., 2014*; *Tang et al., 2015*; *Somervuo et al., 2017*). Here, we detail the construction, curation and evaluation of a database designed for hierarchical classification of amplicon sequences belonging to a 157 bp COI locus commonly used for arthropod metabarcoding (*Zeale et al., 2011*). This work will serve as both a resource for those conducting experiments using arthropod metabarcoding and as a template for future work curating and evaluating hierarchical sequence classification databases.

## METHODS

### Data collection and curation

To produce a comprehensive reference set, all COI annotated sequences from Arthropoda as well as three sister phyla, Mollusca, Onychophora and Tardigrada, between 250 and 2,500 bp in length were downloaded from the NCBI Nucleotide repository on October 21st, 2016 using the search term 'Arthropoda cytochrome oxidase subunit I'. To supplement this collection, all arthropod whole mitochondrion genomes were downloaded from NCBI Nucleotide on March 3rd, 2017 using the search term 'Arthropoda mitochondrion genome'. For metagenetic analysis, the inclusion of close outgroup sequences is useful for estimating the sequence space boundaries between arthropods and alternate phyla. The Perl script provided in *Sickel et al. (2015)* was then used along with the NCBI Taxonomy module (*NCBI Resource Coordinators, 2018*) to retrieve the taxonomic identity of each sequence across each of the major Linnaean ranks, from kingdom to species.

After obtaining the available sequences and rank annotations, we created an intermediate database to obtain extracted barcode amplicons of interest from the reference data using the Metaxa2 database builder tool (v1.0, beta 4; http://microbiology.se/software/metaxa2/). This tool creates the hidden Markov models (HMMs) and BLAST reference databases underpinning the Metaxa2 classification procedures. Prior to extraction, we randomly selected a reference sequence, trimmed it to the exact 157 bp barcode amplicon of interest and designated it as the archetypical reference during database building using the '-r' argument. The section of the arthropod COI gene we trimmed this sequence to is the amplicon product of the commonly used primers of *Zeale et al. (2011)*. The reference sequence is used in the database builder tool to define the range of the barcoding region of interest, and the software then trims the remainder of the input sequences to this region using the Metaxa2 extractor (*Bengtsson-Palme et al., 2015*). To increase the accuracy of multiple sequence alignment during this process, we split the original input sequences on the basis of length prior to running the database builder for amplicon extraction, creating four files with sequences of 250–500 bp, 501–600 bp, 601–2,500 bp and whole mitochondrion genomes. Following sequence extraction, the database builder tool aligns trimmed sequences using MAFFT (*Katoh & Standley, 2013*) and from this alignment the conservation of each residue in the sequence is determined. The most conserved regions are selected for building HMMs using the HMMER package (*Eddy, 2011*). Input sequences that cover most of the barcoding region and are taxonomically annotated are used to build a BLAST (*Altschul et al., 1997*) database for sequence classification. Finally, the sequences

**Table 1 Taxonomic midpoint annotation corrections.** Summary of taxonomic annotations made for references which had undefined ranks at midpoints in their respective taxonomic lineages.

| Undefined rank | Higher resolution assignment | Assignment made | Authority used |
|---|---|---|---|
| Order | Family Sphaerotheriidae | Order Sphaerotheriida | MilliBase |
| Order | Family Zephroniidae | Order Sphaerotheriida | |
| Order | Family Lepidotrichidae | Order Zygentoma | |
| Order | Family Lepismatidae | Order Zygentoma | |
| Order | Family Nicoletiidae | Order Zygentoma | |
| Class | Order Pauropoda | Class Myriapoda | |
| Family | Genus Pseudocellus | Family Ricinididae | ITIS |
| Family | Genus Chanbria | Family Eremobatidae | |
| Genus | Species Tanypodinae spp. | Genus Tanypodinae | |
| Genus | Species Ennominae spp. | Genus Ennominae | |
| Family | Genus Dichelesthiidae | Family Dichelesthiidae | |
| Family | Genus Phallocryptus | Family Thamnocephalidae | |
| Class | Order Symphyla | Class Myriapoda | |
| Order | Family Peripatidae | Order Onychophora | |
| Class | Family Peripatidae | Class Onychophora | *Regier et al. (2010)* |
| Order | Family Peripatopsidae | Order Onychophora | |
| Class | Family Peripatopsidae | Class Onychophora | |
| Family | Genus Lasionectes | Family Speleonectidae | |
| Order | Family Speleonectidae | Order Nectiopoda | |
| Family | Genus Prionodiaptomus | Family Diaptomidae | WoRMS |
| Order | Family Diaptomidae | Order Calanoida | |
| Class | Order Calanoida | Class Maxillopoda | |

in the BLAST database are aligned using MAFFT, and the intra- and inter-taxonomic sequence identities are calculated to derive meaningful sequence identity cutoffs at each taxonomic level. This entire process is described in more detail in the Metaxa2 2.2 manual (http://microbiology.se/software/metaxa2/) and in *Bengtsson-Palme et al. (2018)*.

After extraction, sequences were then curated by removal of duplicate sequences using the Java code provided with the RDP classifier (v2.11; *Wang et al., 2007*), which removes identical sequences or any sequence contained within another sequence. At this point, we conducted extensive curation of the available lineage data for the reference sequence database. For references lacking complete annotation at midpoints within the Linnaean lineage, we used Perl regular expression-based substitution to complete the annotation according to established taxonomic authorities, including MilliBase (*Sierwald, 2017*), the Integrated Taxonomic Information System (http://www.itis.gov) and the phylogenomic analysis of *Regier et al. (2010)*. Table 1 shows the substitutions made. Further, we removed ranks containing annotations reflective of open nomenclature, such as sp., cf. and Incertae sedis, as well as ranks annotated as 'undef.' Lastly, we removed entries containing more than two consecutive uncalled base pairs.

Upon analyzing the representativeness of this initial database across arthropod classes and insect orders, we found that amplicon sequences from two insect orders, Strepsiptera and Embioptera, were not present in the curated database, likely due to their poor sequence similarity to the reference sequence used to designate the amplicon barcode region of interest. To add Strepsiptera and Embioptera COI amplicons, all NCBI COI sequences belonging to these orders were downloaded on October 10th, 2017, curated and added to the Metaxa2 COI database. To improve recovery of amplicons from these insect orders during curation, a representative sequence from both Embioptera and Strepsiptera, representing the 157 bp COI amplicon of interest, was used when building the Metaxa2 database. This retrospective addition of sequences belonging to Strepsiptera and Embioptera contributed 102 and three non-redundant reference sequences to the database, respectively. After this final sequence addition step, a Metaxa2 database was built to include all curated sequences and this database is available through the Metaxa2 software package (http://microbiology.se/software/metaxa2/).

To assess the degree to which our amplicon sequence extraction, dereplication and curation procedures worked, we took inventory of the number of sequences per species in the initial input data as well as the number of sequences and genera present in the data at three points during curation: (1) in the initial input data, (2) following Metaxa2 database builder-based amplicon sequence extraction and (3) in the final database following dereplication and taxonomic curation.

## Classifier performance evaluation

For performance evaluations, the methods used were highly similar to those of *Richardson, Bengtsson-Palme & Johnson (2017)*. For three repeated samplings, we randomly selected 10 percent of the curated sequences to obtain testing data, using the remaining 90 percent of sequences to train the Metaxa2 classifier for performance evaluations. To assess the effect of sequence length on classifier performance, we used a Python script to crop the test case sequences to 80 bp in length, approximately half the median length of the original reference sequence dataset. Evaluating classification performance on these short sequences provides a test of the classifiers robustness to sequence length variation and enables estimation of the potential for classifying sequences from short, high-throughput technology, such as 100 cycle single-end Illumina HiSeq sequencing. We then performed the following analyses on both the full-length (157 bp) and half-length (80 bp) test case sequences, separately.

To characterize the relationship between the Metaxa2 reliability score, an estimate of classification confidence, and the probability of classification error, we used the COI trained classifier to classify the testing datasets, requiring the software to classify to the family rank regardless of the reliability score of the assignment. After comparing the known taxonomic identity of each reference test case to the Metaxa2 predicted taxonomic identity, we regressed 5,000 randomly chosen binary classification outcomes, '1' representing an incorrect classification and '0' representing a correct classification, against the Metaxa2 reliability score using local polynomial logistic regression in R (v3.3.1; *R Core Team, 2014*) with the span set to a value of 0.5.

For each of the three testing and training datasets, we classified the testing sequences using Metaxa2 with a reliability score threshold (-R) of 68. With the resulting classifications, we compared the known taxonomic identity of each reference test case to its Metaxa2 classification, from kingdom to species, to assess the proportion of true positive, true negative, false negative and false positive predictions. We also calculated false discovery rate, as measured in errors per assignment for each rank.

To assess the rate of taxonomic overclassification at the genus and species levels, we searched the testing dataset for sequence cases belonging to arthropod genera and species not represented in the training database. For each of the two ranks, we then determined the proportion of these sequences which were classified. Since the actual identity of such a sequence case is not represented in the training data, any such classification represents a particular type of misclassification known as an overclassification or overprediction. Lastly, for these sequence cases, we looked at how the classifier performed at the preceding rank (e.g., for the species-level cases, we analyzed classifier performance at the genus-level). Such analysis provides a measure of how well the software is able to perform at the next higher rank for these worst-case-scenario input sequences.

For each order in the testing data, we estimated the family, genus and species-level proportion of sequences assigned and false discovery rate to estimate the degree of variance in performance across major arthropod lineages. For this analysis, the false discovery rate was again defined as the number of errors per assignment. After conducting this analysis, we limited our interpretation of the results to orders with at least 100 tests sequence cases at all of the ranks analyzed, family, genus and species. A Python script which takes the testing sequence taxonomies, training sequence taxonomies and Metaxa2 predicted taxonomies as input and provides the summaries of classification performance described above is provided with the GitHub repository associated with this work (https://github.com/RTRichar/Zeale_COI_Database). This work was performed using Ohio Supercomputer Center resources (*Ohio Supercomputer Center, 1987*).

## RESULTS

Following curation and extraction, we obtained 199,206 reference amplicon sequences belonging to 51,416 arthropod species. Over 90 percent of the references were between 142 and 149 bp in length, with a minimum reference sequence length of 94 bp. For the final database creation and classifier training procedure, many reference amplicons were shorter than the 157 bp region of interest due to the incompleteness of some reference sequences and the trimming of taxonomically uninformative ends during Metaxa2 training. Prior to this step, 82 percent of the sequences were between 150 and 157 bp in length following the original extraction and these longer sequences can be found at the GitHub repository associated with this work. The taxonomic representativeness of the database across different arthropod classes and insect orders, including the number of families, genera and species in each, are presented in Tables 2 and 3.

Analyzing the number of sequences per species in the input reference sequence data, we observed a heavily right-skewed distribution, with a median of 2 and a mean of 11.1

**Table 2 Summary of database completeness with respect to all arthropods.** Summary of taxonomic representation across all arthropod classes and associated sister groups. Numbers may include sub and super groupings.

| Class | Number of orders | Number of families | Number of genera | Number of species |
|---|---|---|---|---|
| Heterotardigrada | 1 | 2 | 2 | 1 |
| Eutardigrada | 1 | 3 | 12 | 20 |
| Onychophora | 1 | 2 | 17 | 42 |
| Pycnogonida | 1 | 10 | 27 | 89 |
| Cephalopoda | 1 | 1 | 1 | 1 |
| Merostomata | 1 | 1 | 3 | 4 |
| Arachnida | 17 | 226 | 740 | 1,804 |
| Myriapoda | 2 | 4 | 7 | 9 |
| Chilopoda | 5 | 16 | 53 | 172 |
| Diplopoda | 11 | 33 | 95 | 181 |
| Ostracoda | 2 | 6 | 19 | 40 |
| Branchiopoda | 3 | 25 | 76 | 254 |
| Malacostraca | 13 | 256 | 969 | 2,654 |
| Maxillopoda | 11 | 85 | 240 | 568 |
| Cephalocarida | 1 | 1 | 2 | 1 |
| Remipedia | 1 | 2 | 5 | 8 |
| Protura | 1 | 4 | 12 | 13 |
| Diplura | 1 | 5 | 7 | 11 |
| Collembola | 4 | 18 | 98 | 203 |
| Insecta | 29 | 789 | 14,654 | 45,341 |
| **Total** | **107** | **1,489** | **17,039** | **51,416** |

sequences per species (Fig. 1A). Further, 32.0 percent of species were represented by 5 or more sequences and 40 species, including *Bemisia tabaci* and *Delia platura*, were represented by between 1,000 and 9,736 entries. After conducting amplicon sequence extraction using the Metaxa2 database builder tool, we were able to extract the COI region of interest from 80.8 percent of the input sequences, which corresponded to 73.4 percent of the genus-level diversity found in the original input data. Following sequence dereplication, removal of sequences with three or more ambiguous base calls and taxonomic lineage curation, our final database contained approximately 13 percent of the input extracted sequences, which represented 98.2 percent of the genus-level richness of the input extracted reference amplicon sequences (Figs. 1B and 1C).

Regressing classification outcome against the Metaxa2 reliability score yielded a similar best fit model for both the 80 bp and full length test sequence datasets (Fig. 2). For both regressions, the probability of sequence mis-assignment was below 10 percent for reliability scores above 70. Thus, for our evaluations, we chose a reliability score of 68, which corresponded to family-level error probabilities of approximately 11.3 percent and 9.5 percent for 80 bp and full length sequences, respectively.

In evaluating the performance of our classification database when analyzed with Metaxa2 at a reliability score cutoff of 68, we found consistently low proportions of false positives

**Table 3  Summary of database completeness with respect to insects.** Summary of insect taxa included in the arthropod COI database following curation. Numbers may include sub and super groupings.

| Order | Number of families | Number of genera | Number of species |
|---|---|---|---|
| Archaeognatha | 2 | 15 | 18 |
| Zygentoma | 3 | 3 | 3 |
| Odonata | 34 | 243 | 488 |
| Ephemeroptera | 26 | 108 | 378 |
| Zoraptera | 1 | 1 | 1 |
| Dermaptera | 4 | 6 | 7 |
| Plecoptera | 15 | 84 | 199 |
| Orthoptera | 30 | 299 | 603 |
| Mantophasmatodea | 1 | 1 | 1 |
| Grylloblattodea | 1 | 1 | 0 |
| Embioptera | 1 | 2 | 2 |
| Phasmatodea | 6 | 20 | 28 |
| Mantodea | 11 | 74 | 76 |
| Blattodea | 8 | 82 | 95 |
| Isoptera | 6 | 91 | 186 |
| Thysanoptera | 3 | 14 | 30 |
| Hemiptera | 101 | 1,245 | 2,730 |
| Psocoptera | 12 | 17 | 19 |
| Hymenoptera | 74 | 1,500 | 4,418 |
| Raphidioptera | 2 | 9 | 11 |
| Megaloptera | 2 | 10 | 22 |
| Neuroptera | 14 | 65 | 153 |
| Strepsiptera | 3 | 10 | 36 |
| Coleoptera | 124 | 2,287 | 7,452 |
| Trichoptera | 43 | 320 | 1,269 |
| Lepidoptera | 134 | 6,554 | 21,626 |
| Siphonaptera | 6 | 14 | 20 |
| Mecoptera | 4 | 5 | 10 |
| Diptera | 118 | 1,574 | 5,460 |
| **Total** | **789** | **14,654** | **45,341** |

across all ranks, though the proportion of true positive, true negative and false negatives varied more considerably from kingdom to species (Fig. 3A). Further, while the Metaxa2 false discovery rate increased with higher resolution ranks (Fig. 3B), it was generally low, never exceeding 5 percent at the genus level. Interestingly, Metaxa2 displayed low variance in the proportion of sequences assigned when classifying 80 bp sequences relative to full length sequences of 147 bp in median length. Overall, the proportion of sequences assigned was greater than 90 percent through the order level for both full length and half length sequences. Beyond the order level, this statistic decreased to 53 and 56 percent at the species level for half length and full length sequences, respectively. Conversely, the proportion of false positives varied more strongly by sequence length and was greatest at higher-resolution

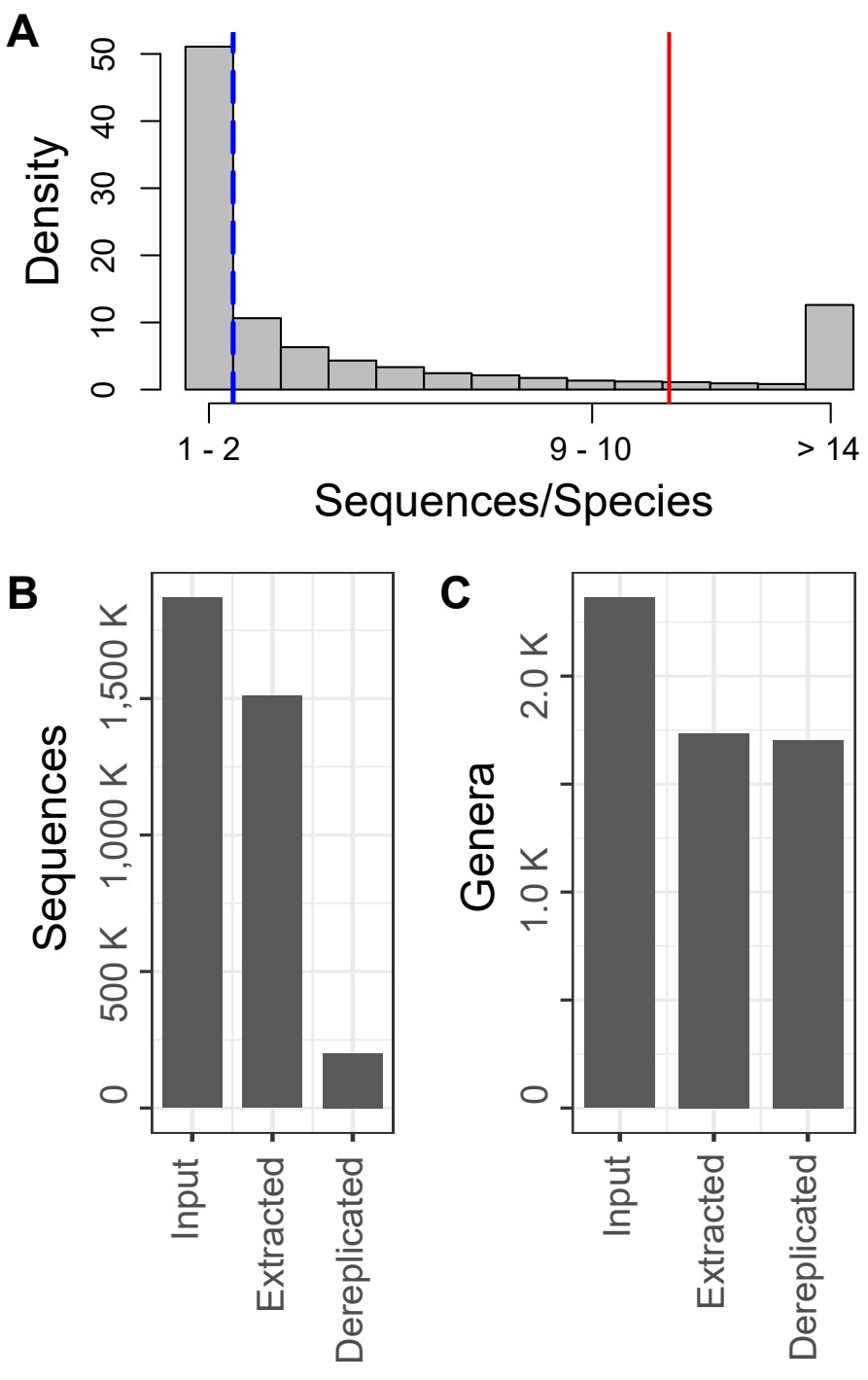

**Figure 1   Data inventory throughout curation process.** A percent density histogram of the number of sequences per species (A) shows the distribution of redundancy within the NCBI Nucleotide entries used. The dashed blue line and solid red line indicate the median and mean number of sequences per species, respectively. Inventories of the number of sequences (A) and genera (B) input into the curation process, following Metaxa2 extraction and following dereplication of redundant sequences and curation of taxonomic lineages.

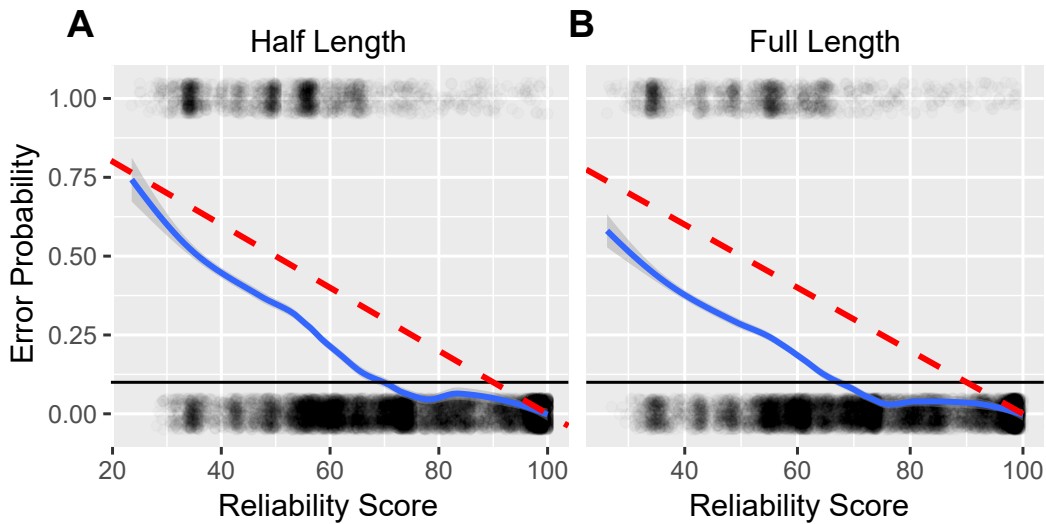

**Figure 2** **Estimating reliability score accuracy.** A logistic regression analysis of case-by-case classification accuracy, '1' indicating a false-positive identification and '0' indicating a true-positive identification, regressed against classification reliability score for half length (A) and full length (B) test sequence cases. A best fit local polynomial regression line (solid blue with 95 percent confidence interval) was used to estimate the relationship between reliability score and the probability of mis-classification. Dashed red lines illustrate the hypothetically ideal 1 to 1 relationship between error probability and the Metaxa2 reliability score, an estimate of classification confidence. Solid black lines highlight the 10 percent error probability.

taxonomic levels. At the species level, 1.97 percent of 80 bp sequences were misclassified, compared to only 1.13 percent for full length sequences. At the order level, the percent of sequences misclassified was 0.59 and 0.65 for 80 bp and full length sequences, respectively. As measured in errors per assignment, the classification false discovery rate was similarly highest at the species level, with 7.3 and 6.3 percent of assignments being incorrect for 80 bp and full length sequences, respectively. False discovery rates decreased to 1.2 and 0.7 percent of assignments being incorrect at the order level for 80 bp and full length sequences, respectively.

During our evaluation of taxonomic overclassifiction, we found between 3,141 and 3,202 sequence test cases belonging to species not represented in the corresponding training data and between 612 and 630 sequence test cases belonging to genera not represented in the corresponding training data across the three iterations of training and testing data. At the species level, the proportion of these cases which were overclassified was roughly equal for full length and half length sequences, with 5.4 and 5.2 percent being overclassified (Fig. 4A). With respect to genus level classification performance on these species overclassification test cases, approximately 31 and 37 percent of test cases were classified correctly as true positives or true negatives for half length and full length sequences. Genus level false positive proportions for these sequence cases were 6.2 and 5.1 percent (Fig. 4B). For the genus level overclassification cases, the difference in overclassification rates by sequence length was slightly larger, with approximately 8.9 and 10.9 percent of full length and half length test cases being overclassified (Fig. 4C). Family level performance on genus level

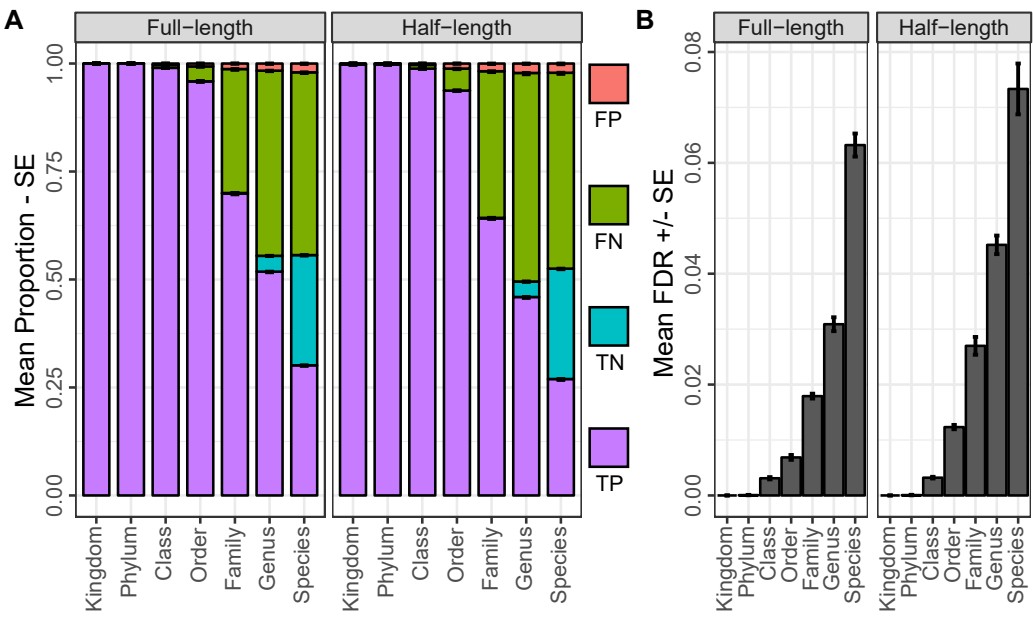

**Figure 3 Classification performance accross entire testing corpus.** Mean proportion and standard error of true positives (TP), true negatives (TN), false negatives (FN) and false positives (FP) for the classification of all testing sequences, conducted on both the full-length and half-length sequences (A). Mean and standard error of the false discovery rate for both full length and half length sequences, as measured in errors per assignment, during classification of all testing sequences (B).

overclassification cases was slightly lower with the proportion of true positive and true negative identifications summing to 25 and 27 percent for half length and full length sequences, with corresponding false positive proportions of 8.8 and 8.0

Evaluating classification performance for each order resulted in unsurprising outcomes across taxonomic ranks, from family to species (Fig. 5). Generally, the proportion of sequences assigned decreased and false discovery rate increased with increasing taxonomic resolution. Between orders, there was noteworthy variation in performance. For example, the proportion of sequences assigned was highest among trichopteran sequences and lowest among lepidopteran sequences, with approximately 94 and 50 percent of sequences being assigned to the family rank for each order, respectively. Further, while the proportion of sequences assigned was similarly lowest for lepidopteran sequences at the genus and species levels, wherein 39 and 21 percent of sequences were assigned for each respective rank, the highest proportion of sequences assigned at these ranks was not observed with trichopteran sequences. Instead, genus and species level proportion assigned was highest among Sessilia sequences, with 86 and 78 percent of sequences being assigned to each respective rank.

## DISCUSSION

While species-specific PCR and immunohistochemistry-based methods have been useful in documenting arthropod food webs (*Stuart & Greenstone, 1990*; *Symondson, 2002*; *Weber et al., 2006*; *Blubaugh et al., 2016*), the narrow species-by-species nature of such approaches

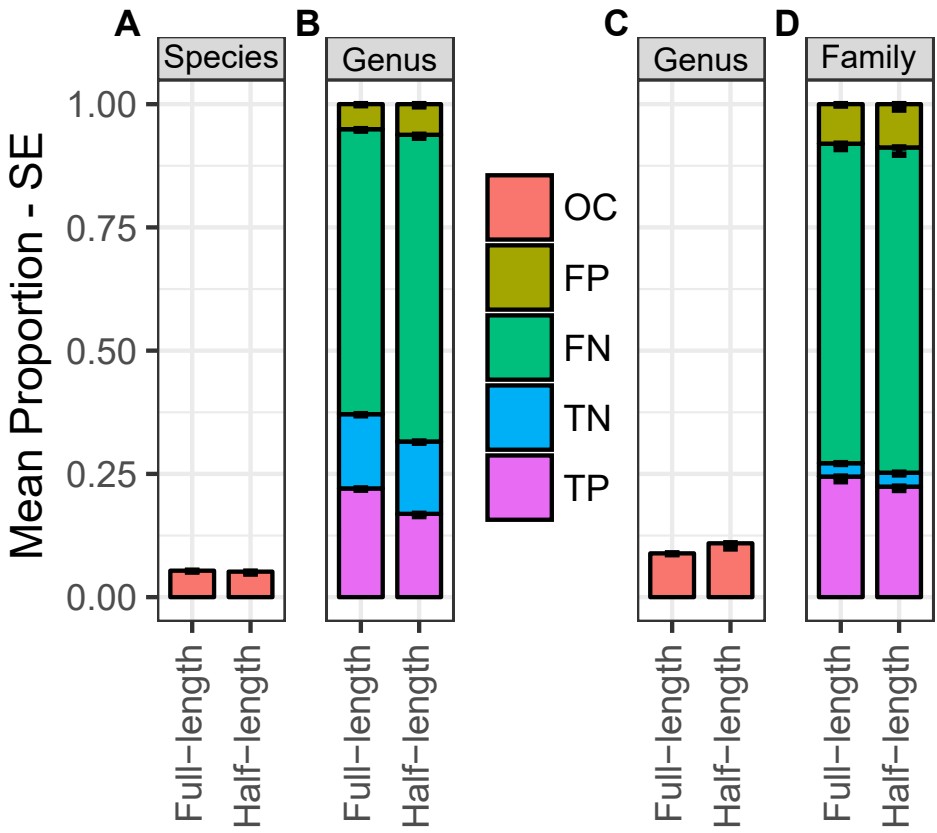

**Figure 4 Overclassification analysis.** Proportional species level overclassification rate (A) and genus level classification performance (B) for test sequence cases from species not represented in the corresponding training data. Proportional genus level overclassification rate (C), and family level classification performance (D) for test sequence cases from genera not represented in the corresponding training data.

has limited their utility for answering large-scale or open-ended ecological questions. With the increasing availability of high-throughput sequencing, arthropod metabarcoding will continue to become more broadly applicable to scientific questions spanning a diversity of research areas. The development of improved methods for drawing maximal inferences from sequence data is an important area for further methodological research. In creating a highly curated COI reference amplicon sequence database and evaluating its performance when used with the Metaxa2 taxonomic classifier, we have developed a new method to aid researchers in the analysis of arthropod metabarcoding data.

Though predictions vary greatly, researchers have estimated the species richness of arthropods to be between 2.5 to 3.7 million (*Hamilton et al., 2010*). Further, according to the literature review of *Porter et al. (2014)*, 72,618 insect genera have been described to date. Thus, the 51,416 species and 17,039 genera represented in our database account for only a small fraction of arthropod biodiversity. The limited representativeness of currently available, high quality reference sequence amplicons for the COI region highlights the

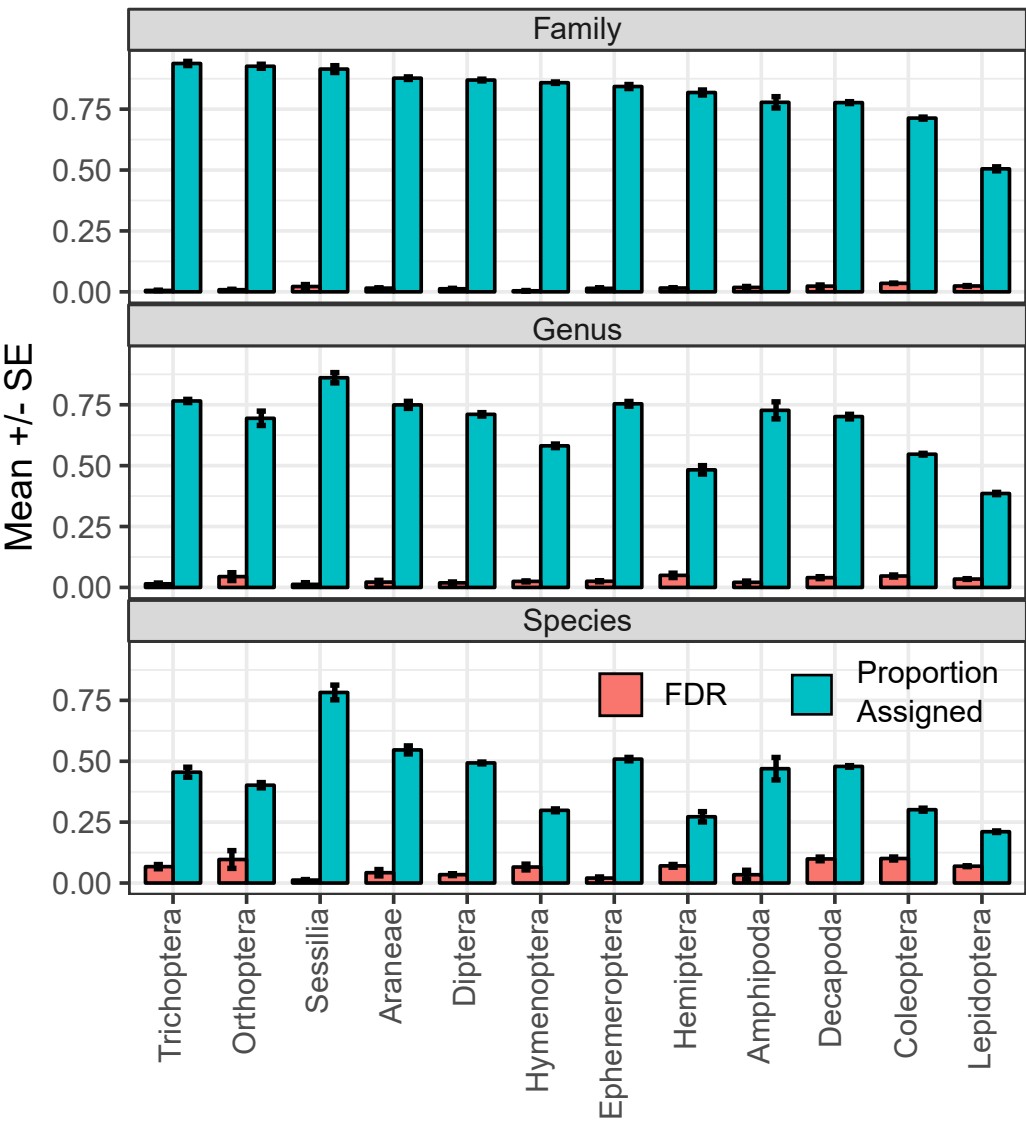

**Figure 5   Classification performance by order.** Mean and standard error of the proportion of sequences assigned and false discovery rate, as measured in errors per assignment, measured at the family, genus and species levels for sequences belonging to each order. These results represent classification performance by arthropod order for the full length sequences only and orders with fewer than 100 test sequence cases at any of the ranks analyzed were excluded from analysis.

need for continued efforts to catalogue arthropod biodiversity with molecular techniques. Despite this current limitation, the combination of molecular gut content analysis with high-throughput sequencing is a promising path toward investigating arthropod trophic ecology and biodiversity monitoring with greater sensitivity and accuracy relative to alternate approaches.

The results of our inventory of sequences per species and genus level richness at various stages in the database curation process revealed that our amplicon extraction procedure

was highly sensitive, extracting and trimming approximately 81 percent of the input sequences down to the 157 bp region of interest. Further, approximately 87 percent of these extracted sequences represented sequence redundancies and were removed during dereplication. As mentioned previously, the trimming of sequence residues adjacent to the barcode of interest and removal of redundant sequences not only makes computational analysis less resource intensive, it can also improve classification performance. For k-mer style classifiers, extraneous sequence residues can bias model selection during classifier training, while abundant sequence duplicates can result in an overwhelming number of identical top hit alignments for alignment-based classifiers.

Overall, the best fit local regression models summarizing the relationship between the Metaxa2 reliability score and the probability of classification error were useful in that the likelihood of misclassification was always less than what would be expected based on the reliability score. For example, a reliability score of 90 corresponded to only a 3.3 percent probability of family-level misclassification for full length sequences. We selected a reliability score of 68 for subsequent analysis as this provided a balanced trade-off between sensitivity and accuracy. Using this reliability score we observed minimal false positive rates and overall proportions of misclassification when comparing our results to those of similar studies (*Porter et al., 2014*; *Bengtsson-Palme et al., 2015*; *Edgar, 2016*; *Richardson, Bengtsson-Palme & Johnson, 2017*). Given that the family level probability of error was only 9.5 percent at a reliability score of 68, a lower reliability score threshold may be justifiable for certain research situations. However, further testing should be conducted to ensure that the relationship between reliability score and classification confidence is similar across taxonomic ranks and between different DNA barcodes.

With respect to sensitivity using a reliability score of 68, our results were highly dependent upon the rank being analyzed, with sensitivities, as measured by the total proportion of sequences classified, above 60 percent only being achieved at the family and order ranks. To some degree, these sensitivity estimates reflect the large degree of database incompleteness at the genus and species ranks, wherein approximately 3.6 and 25.5 percent of unclassified sequences were true negatives. However, to our knowledge, no other studies have reported classification sensitivity data for this COI amplicon locus. This makes it difficult to ascertain if Metaxa2 –with an average of approximately 44.7 percent of false negative assignments at the genus and species ranks –exhibits relatively low sensitivity or if this locus is limited in discriminatory power. The short length of the 157 bp COI amplicon region relative to other barcoding regions such as the ITS and 18S rRNA regions (*Hugerth et al., 2014*; *Wang et al., 2015*) could be a cause of such limited discriminatory power.

As expected, analyzing cases of overclassification in our data revealed that sequences from taxa lacking representation in the database are far more likely to be misclassified relative to sequences from well-represented taxa. This is supported by an approximately 10 percent probability of genus level overclassification for sequences from unrepresented genera relative to a 1 to 2 percent probability, depending on sequence length, for all sequence test cases. Interestingly, the genus level overclassification rate was approximately double that observed at the species level. This seems counterintuitive but is expected in light of the discussion put forth by *Edgar (2018)*, wherein the percent identity difference from

the closest reference sequence match is considered one of the most important predictors of classification performance. For a species-level overclassification case, the closest reference sequence match in the corresponding database is a sequence from a congeneric species in the best case scenario. Since the closest reference sequence match for a genus level overclassification case is a sequence from a confamilial species at best, the average percent identity difference from the closest reference is greater for a genus level overclassification case than for a species level overclassification case. Thus, for overclassification cases in particular, higher levels of error are expected at more inclusive taxonomic ranks.

While the genus level overclassification estimates we observed are not desirable, they are considerably lower than similar estimates for the RDP classifier, which range from 21.3 percent to 67.8 percent depending on the database, locus analyzed and cross-validation approach used (*Edgar, 2016*; *Richardson, Bengtsson-Palme & Johnson, 2017*). Further, the observed degree of genus level overclassification using Metaxa2 with our COI database was similar to or less than that of the recently developed SINTAX classifier (*Edgar, 2016*; *Edgar, 2018*). Interestingly, the recent analysis of *Edgar (2018)*, resulted in a similar estimate of the Metaxa2 genus level overclassification rate while revealing a considerably higher overclassification rate for the majority of other classifiers tested. Though this analysis found Metaxa2 to be relatively less sensitive than other classifiers, a weakness of the work was that it only tested Metaxa2 using the default reliability score of 80, while testing multiple confidence thresholds for alternate classifiers, such as SINTAX and RDP. Given that our analysis has revealed the Metaxa2 default reliability score to be too conservative—at least for the COI locus—such results are difficult to interpret. In general, such comparisons across studies should be approached with caution as multiple factors complicate the interpretation of classification performance, such as locus discriminatory power, database completeness and the choice of evaluation metrics used. Ultimately, direct comparisons of classification methods using standardized loci and databases are needed to more rigorously compare performance.

With respect to Metaxa2 classification of full length relative to half length amplicon sequences, we observed surprisingly small differences in performance. Consistently, the proportion of misclassified sequences was greater for half length sequences. When considering error and sensitivity together, the false discovery rate or errors per assignment for full length sequences was consistently less than or equal to that achieved during the classification of half length sequences. Lastly, when considering the relationship between the Metaxa2 reliability score and the probability of classification error at the family level, we noted highly similar local polynomial regression models of error probability for both full length and half length sequences.

## CONCLUSIONS

Here, we assembled a highly curated database of arthropod COI reference amplicon sequences, trained a recently developed hierarchical DNA sequence classifier using the database and conducted extensive *in silico* performance evaluations on the resulting classification pipeline. Overall, we found a high degree of sequence redundancy within the

initial, uncurated dataset, highlighting the importance of effective sequence dereplication during the creation of databases designed for metabarcoding analysis. Further, the limited representativeness of the database with respect to arthropod biodiversity indicates that additional sequencing effort is needed to further improve the performance of arthropod metabarcoding techniques. Though the performance evaluations presented in this work were conducted on a large corpus of available biological data, the results are not necessarily directly transferable to all experimental settings. For example, variations in sequence error profiles and taxonomic distributions among datasets are potential confounding factors. Despite these limitations, this work provides researchers with a new resource for arthropod COI sequence analysis and novel data for gauging the strengths and limitations of different approaches to arthropod metabarcoding.

## ACKNOWLEDGEMENTS

This work was supported by an allocation of computing time from the Ohio Supercomputer Center.

### Funding

This work was supported by a Project Apis m.—Costco Honey Bee Biology Fellowship to Rodney Richardson, and state and federal appropriations to the Ohio Agricultural Research and Development Center (USDA-NIFA Projects OHO01277 and OHO01355-MRF). Johan Bengtsson-Palme was supported by the Swedish Research Council for Environment, Agricultural Sciences and Spatial Planning (FORMAS; grant 2016-00768). The funders had no role in study design, data collection and analysis, decision to publish, or preparation of the manuscript.

### Grant Disclosures

The following grant information was disclosed by the authors:
Costco Honey Bee Biology Fellowship.
Ohio Agricultural Research and Development Center: OHO01277, OHO01355-MRF.
Swedish Research Council for Environment, Agricultural Sciences and Spatial Planning: 2016-00768.

### Competing Interests

The authors declare there are no competing interests.

### Author Contributions

- Rodney T. Richardson conceived and designed the experiments, performed the experiments, analyzed the data, contributed reagents/materials/analysis tools, prepared figures and/or tables, authored or reviewed drafts of the paper, approved the final draft.
- Johan Bengtsson-Palme performed the experiments, contributed reagents/materials/-analysis tools, authored or reviewed drafts of the paper, approved the final draft.

- Mary M. Gardiner and Reed M. Johnson conceived and designed the experiments, authored or reviewed drafts of the paper, approved the final draft.

**Data Availability**

The entire set of curated sequences associated with this work can be downloaded from the associated GitHub repository: https://github.com/RTRichar/Zeale_COI_Database. This repository has additionally been archived using Zenodo (DOI: 10.5281/zenodo.1256869).

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
