# Peer review of "A reference cytochrome c oxidase subunit I database curated for hierarchical classification of arthropod metabarcoding data"

_PeerJ, doi:10.7717/peerj.5126_

## Round 0.1 · original submission · Minor Revisions

Dear Dr. Richardson and colleagues:

Thanks for submitting your manuscript to PeerJ. I have now received three independent reviews of your work, and as you will see, the reviewers raised some concerns about the research. In particular, please consider comments about experimental design, statistics, explanation of protocols and approaches, and improvements to overall presentation. Also make sure that all terms are defined, and that all methods are clearly explained. Therefore, I am recommending that you revise your manuscript accordingly, taking into account all of the issues raised by the reviewers. I do believe that your manuscript will be ready for publication once these issues are addressed.

Good luck with your revision,

-joe

Reviewer 1 ·

Basic reporting

- The manuscript is written in good English and is well understandable.
- The use of references/literature is appropriate and sufficient background/context is provided
-The manuscript is well structured and raw data is available.
-Results appropriatley show the performance of the used software/pipeline.

Experimental design

- The manuscript fits the aims and scope of the journal
- The reason for developing the described pipeline for taxonomic assignemnt and the need for this kind of research becomes clear.
- The described pipeline has been sufficiently tested and methods are described with sufficient detail.

Validity of the findings

-The findings are valid and the benefits of using the described approach/pipeline for building reference databases are evident.
-Conclusions are well stated and support the results.

·

Basic reporting

1. l. 92 Sickel et al. 2015
2. l. 93 Why do you cite Sayers et al. 2013 specifically? The "Database Resources of the National Center for Biotechnology Information" is published yearly with multiple newer iterations, e.g. https://doi.org/10.1093/nar/gkx1095 and https://doi.org/10.1093/nar/gkw1071
3. l. 138-142 This is a good way to make sure that there are no other taxonomic groups that are lost for similar reasons as Strepsiptera and Embioptera. Other groups might be lacking not entirely but in large fractions this analysis should reveal them. I can't find the results of this inventory at the three stages anywhere in the paper. It would be nice to present them in some way (at least as supplementary tables).
4. l. 222 Figure 3B

Experimental design

1. l. 87 Why use 2500bp as the cutoff?
2. l. 88 What was the exact search term used?
3. l. 99 How was this archetypical sequence selected? Is it an actual sequence or a reconstructed ancient sequence?
4. l. 104 How does splitting the original sequences into multiple files increase the accuracy of this process?
5. l. 160 Why use 5000 instead of all (~20000) binary classification outcomes?

Validity of the findings

no comment

Additional comments

The authors describe the construction and evaluation of a COI database for hierarchical classification.
The construction and curation process is well documented and reproducible.
The evaluation of the classification performance of this database with metaxa2 is thorough.
In combination this work is useful for researchers in need of a COI classification database, as well as researchers wanting to create, curate, or evaluate a database themselves.
I raised some minor points that I would like to see addressed by the authors above.
In addition I have the following general comments:
1. It is a nice idea to publish the data and code via Github. You could consider additionally archiving it via Zenodo https://guides.github.com/activities/citable-code/ This way it will be available in the exact version with a unique doi independent of Github.
2. Using a single split into test/training set gives limited information about the robustness of estimates. The whole process could be repeated multiple times.
3. Does the final data included in metaxa2 also contain the test data?

Reviewer 3 ·

Basic reporting

no comment

Experimental design

no comment

Validity of the findings

no comment

Additional comments

This is a very nice paper, generally well-written, that describes how the authors constructed a database that can be used for taxon identification when using the Zeale primer set in a metabarcoding study. The authors are correct that there are many technical aspects of metabarcoding and other molecular methods using NGS platforms that need to be explored, and this is a contribution toward constructing a better reference database.

1. I understand that the extensive curation resulted in a more compact database that would result in faster processing times, but it is not entirely clear what is gained by the extraction and trimming and removal of redundancy to create the curated dataset. How much is processing time reduced? How much better is the true positive rate? As COI sequences and whole mitogenomes are being added to public databases at an ever increasing rate, how often will the database need to be recurated? How easy will it be to recurate it? The authors should consider addressing some of these questions in the present paper.
2. On a different tack, it would be very interesting if the authors could show (or not) that the accuracy of classification of test sequences was related to the taxonomic “density” of training sequences. For example, is classification of an unknown Lepidopteran more accurate than for an unknown beetle? What about Hemiptera compared to Orthoptera? Also the comparison can be made at the family level, and could be done more generally if “density” were appropriately quantified. This would be very useful for people to know.
3. The authors should acknowledge that theirs is an in silico analysis, and suggest ways in which their conclusions could be verified with true unknowns, or with new sequences.
4. Sensitivity is not defined (at least I could not find a definition). As this is one of the most used words in the results and discussion, it is critical to have it clearly defined. Technically, sensitivity is the true positive rate, and I assumed that it is used this way, and it would be very helpful to state this in the methods. However, in Figure 3C, family level sensitivity is only .2 to .3. If sensitivity is true positive rate, these values seem extremely low. Also, at line 260, the authors suggest a tradeoff between sensitivity and accuracy. However, if the sensitivity is true positive rate, then how does this differ from accuracy? In addition, false positive rate plus true positive rate = 1, so it doesn’t make sense to say there is a trade-off between these. This leaves me wondering what is sensitivity?
5. A statistical technicality, that would be a great improvement (although perhaps not critical here), would be to use cross-validation to calculate true and false positive rates. By splitting the data set only once (or possibly twice – this is not clear to me), the best estimates of these rates are not obtained. However, implementing cross-validation on the dataset would not be a simple task.
6. I understand from the stated methods that all of the Bemisia tabaci sequences were boiled down to one sequence in the final reference database. However, several researchers have suggested that the Bemisia biotypes may be species, so it seems that the curated database should include this variation. Also, other examples with extreme “intra-specific” variation should be similarly addressed.

Minor points easily addressed
Line 266. The statement also applies to different amplicons of the COI barcode, not just different barcode loci.
Lines 275ff. It seems strange to refer to a part of a gene as a locus, and then to refer to genes as regions. I think it would be better to refer to the region of the COI locus coding for the Zeale amplicon, so that COI and 18S can be referred to as loci.
Line 277-279. The use of test sequences from genera that are not in the training dataset is very nice, but it does not distinguish unrepresented taxa from well-represented taxa. It indicates that classification is more accurate for represented genera than for unrepresented genera. My point 2 above could address the effect (if any) of well-represented taxa.
Fig 1 legend. There is no reference to panel C
It is not clear if the github Zeale file is the curated dataset or the longer sequences between 150 and 157 bp (line 179-181). I was not able to see the file contents.

---

## Round 0.2 · accepted · Accept

Dear Dr. Richardson and colleagues:

Thanks for revising your manuscript based on the minor concerns raised by the reviewers. I now believe that your manuscript is suitable for publication. Congratulations! I look forward to seeing this work in print, and I anticipate it being an important resource for the Entomology, Evolutionary Biology and Population Biology fields. Thanks again for choosing PeerJ to publish such important work.

Best,

-joe

#